

# The role of sirtuin1 in liver injury: molecular mechanisms and novel therapeutic target

Mufei Wang[1,2,*], Juanjuan Zhao[3,*], Jiuxia Chen[1], Teng Long[1], Mengwei Xu[1], Tingting Luo[1], Qingya Che[1], Yihuai He[2] and Delin Xu[1]

[1] Department of Medical Instrumental Analysis, Zunyi Medical University, Zunyi, Guizhou, China
[2] Department of Infectious Diseases, Affiliated Hospital of Zunyi Medical University, Zunyi, Guizhou, China
[3] Department of Immunology, Zunyi Medical University, Zunyi, Guizhou, China
[*] These authors contributed equally to this work.

## ABSTRACT

Liver disease is a common and serious threat to human health. The progression of liver diseases is influenced by many physiologic processes, including oxidative stress, inflammation, bile acid metabolism, and autophagy. Various factors lead to the dysfunction of these processes and basing on the different pathogeny, pathology, clinical manifestation, and pathogenesis, liver diseases are grouped into different categories. Specifically, Sirtuin1 (SIRT1), a member of the sirtuin protein family, has been extensively studied in the context of liver injury in recent years and are confirmed the significant role in liver disease. SIRT1 has been found to play a critical role in regulating key processes in liver injury. Further, SIRT1 seems to cause divers outcomes in different types of liver diseases. Recent studies have showed some therapeutic strategies involving modulating SIRT1, which may bring a novel therapeutic target. To elucidate the mechanisms underlying the role of sirtuin1 in liver injury and its potentiality as a therapeutic target, this review outlines the key signaling pathways associated with sirtuin1 and liver injury, and discusses recent advances in therapeutic strategies targeting sirtuin1 in liver diseases.

## INTRODUCTION

The liver plays a critical role in maintaining human health by carrying out a multitude of physiological processes. Unfortunately, liver disease is a common and serious threat to human health caused by a range of factors like chemical exposure, viral infections, underlying diseases, genetic factors (*Zhang et al., 2018*). The pathology of liver disease is both intricate and multifaceted, with hepatocyte function and morphology being impacted by changes in oxidative stress, inflammation, bile acid (BA) metabolism, and autophagy, all playing important roles (*Rovegno et al., 2019*). Based on the different pathogeny, pathology, clinical manifestation, and pathogenesis, liver diseases are grouped into different categories. In this manuscript, we summarized the role of SIRT1 in acute liver injury (ALI), alcoholic liver disease (ALD), nonalcoholic fatty liver disease (NAFLD), ischemia-reperfusion injury

Corresponding authors
Yihuai He, 993565989@qq.com
Delin Xu, xudelin2000@163.com

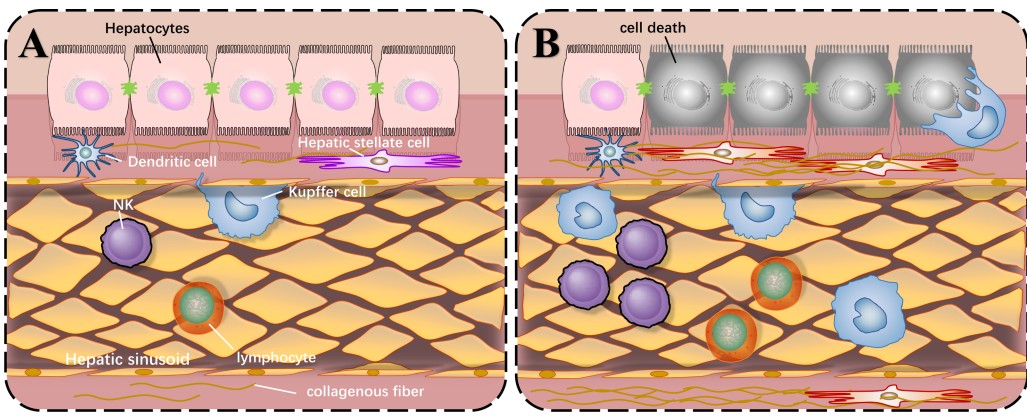

**Figure 1  Local structure diagram of the liver.** (A) Local structure of normal liver; (B) General condition after liver injury.

(IRI), cholestatic injury, and liver fibrosis. However, the mechanisms that cause liver injury are not fully understood, and it is imperative to explore novel treatments for these types of injury.

Sirtuins (SIRTs) belong to the class III histone deacetylase family of proteins are crucial for various biochemical processes within cells, including oxidative stress, inflammation, BA metabolism, and autophagy (*North & Verdin, 2004*). The discovery of Sir2 in budding *Saccharomyces cerevisiae* over four decades ago marked the beginning of extensive research on SIRTs (*Rine et al., 1979*). Of the seven known members of the SIRT family, SIRT1 has been the subject of the most thorough investigation in the context of liver injury and implicated in hepatic metabolism by its ability to deacetylate key metabolic players. Although *Wu et al. (2022)* have conducted a comprehensive review of the SIRT family, the mechanisms that underlie the role of SIRT1 in liver injury are still unclear. A deeper understanding of this molecule in liver injury is critical for the development of effective treatments for liver diseases. This review, in turn, may lead to the development of new and effective therapeutic interventions.

Before we describe the roles of SIRT1 in liver injury, the common local structure diagram of the normal and injured liver is important to understand how liver injury happens. There are various types of immune cells in the normal liver. The liver cells are arranged in a single layer to form a liver plate. The hepatic sinusoids are located between the liver plates. There are colonized Kupffer cells and more NK cells in the hepatic sinusoids, and hepatic stellate cells exist in the perisinusoidal space. When liver injury occurs, liver cells die, and immune cells are activated and proliferated under the stimulation of corresponding antigens. The quiescent hepatic stellate cells are activated and proliferated abnormally, producing extracellular matrix and increasing intrahepatic fibers, which can lead to cirrhosis as shown in Fig. 1.

While liver injuries lead to a same result, apoptosis of hepatocytes, different types of liver injuries caused by various factors represent their own characters. SIRT1 will play distinct

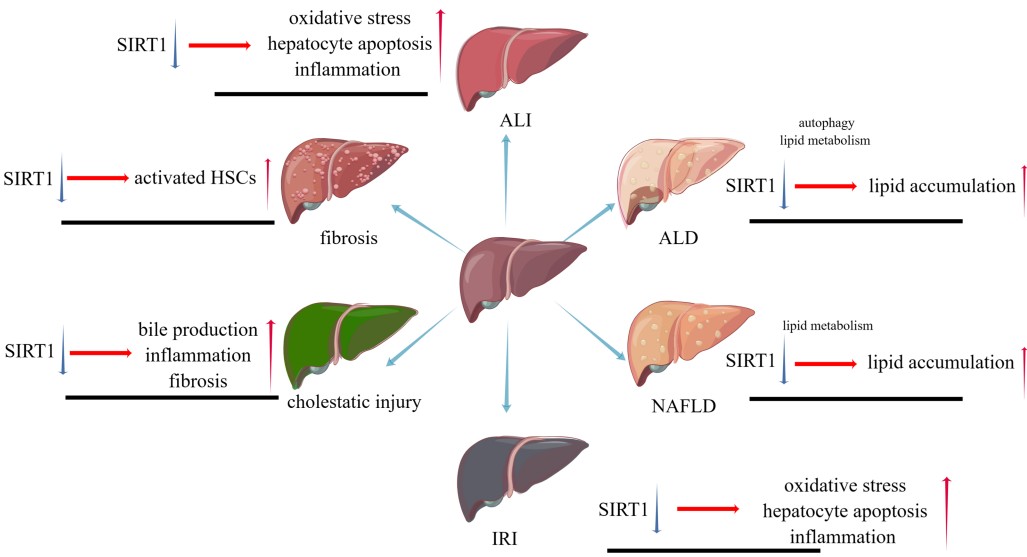

**Figure 2  The pathophysiological changes caused by SIRT1 inhibition in different liver diseases.**

roles in these types, positively or negatively. The pathophysiological changes caused by SIRT1 inhibition in different liver diseases are shown in Fig. 2.

This review will help researchers in molecular biology or hepatic disease fully understand the role of SIRT1 in various liver diseases and the methods of interfering with liver diseases through SIRT1 in recent years.

# SURVEY METHODOLOGY

For harvesting the articles about researches on sirtuin1 in liver injury, the literature research was conducted in the Web of Science, Google Scholar database, and PubMed. We used the following keywords: SIRT1, liver injury, liver fibrosis, liver ischemic, NAFLD, and ALD. Emphasis was placed on articles published since 2018, but earlier articles were also included. Studies included original studies and reviews in English that contained information about SIRT1 in liver disease.

## SIRT1 in the mechanisms of liver injury

The etiology of hepatic injury encompasses intricate mechanisms entailing a myriad of biological processes intrinsic to hepatocytes, several of which remain incompletely understood. Nevertheless, contemporary investigations have underscored the indispensable contribution of SIRT1 in an array of cellular phenomena encompassing inflammation, oxidative stress, and autophagy. Notably, there are distinctive functions of SIRT1 in different cells, which were showed in Figs. 3, 4 and 5. SIRT1 appears to exert disparate effects across distinct hepatic disorders, thereby signifying its nuanced involvement in disease pathogenesis.

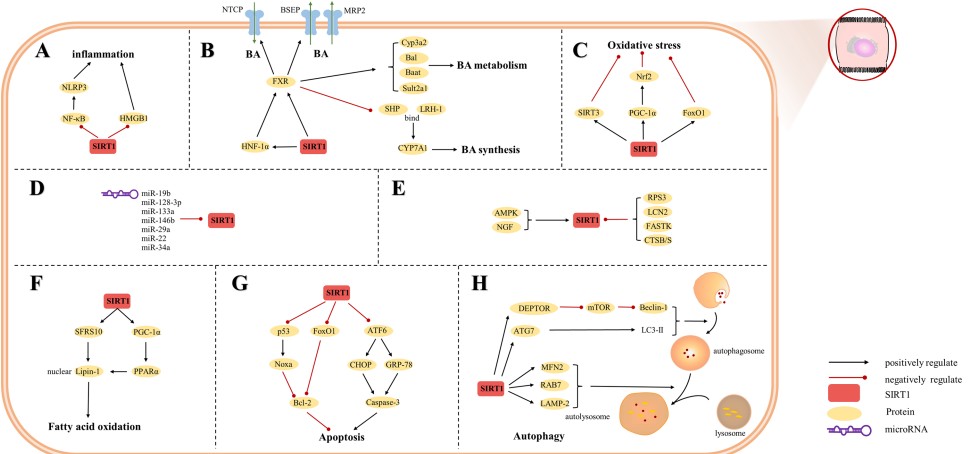

**Figure 3** **Overview of the roles of SIRTs in hepatocytes.** (A) SIRT1 plays an anti-inflammatory effect by regulating inflammatory mediators; (B) SIRT1 is a key factor in BA regulation; (C) SIRT1 plays an anti-oxidative stress role in hepatocytes; (D) MicroRNAs are inhibitors of SIRT1; (E) The upstream proteins of SIRT1; (F) SIRT1 is involved in fatty acid oxidation; (G) SIRT1 reduces hepatocytes apoptosis; (H) autophagy is regulated by SIRT1. (AMPK) adenosine 5′-monophosphate-activated protein kinase, (ATF6) activating transcription factor 6, (ATG7) autophagy-related gene 7, (DEPTOR) DEP domain-containing mTOR-interacting protein, (CHOP) C/EBP-homologous protein, (CTSB/S) cathepsin B/S, (FASTK) fas-activated serine/threonine kinase, (FoxO1) forkhead box-containing protein O 1, (FXR) farnesoid X receptor, (GRP-78) 78-kD glucose-regulated protein, (HMGB1) high mobility group box 1, (HNF-1a) hepatic nuclear receptor-1a, (LAMP-2) lysosomal-associated membrane protein 2, (LCN2) lipocalin-2, (LC3) light chain 3, (LRH-1) liver receptor homolog-1, (MFN2) mitofusin 2, (mTOR) mechanistic target of rapamycin, (NF-$\kappa$ B) nuclear factor-$\kappa$ B, (NGF) nerve growth factor, (NLRP3) NOD-like receptor pyrin domain containing 3, (Nrf2) Nuclear factor erythroid 2-related factor 2, (PGC-1$\alpha$) peroxisome proliferator-activated receptor-gamma coactivator 1-alpha, (PPAR) peroxisome proliferator-activated receptor, (SHP) small heterodimer partner, (SFRS10) serine-rich 10.

### SIRT1 in ALI

Clinically, ALI is defined as jaundice, impaired coagulation function (International Normalized Ratio (INR) >1.5), and more than two times of liver transaminases such as alanine aminotransferase (ALT), aspartate aminotransferase (AST), the markers of hepatocyte damage (*Roohani & Tacke, 2021*), and may further evolved into failure. Diverse factors, including hepatotoxic drugs (*e.g.*, acetaminophen/paracetamol, phenprocoumon, antibiotics, antiepileptics), acute viral hepatitis and infection induce the ALI.

In recent years, the comprehension of ALI mechanism has experienced notable advancements with the elucidation of SIRT1. The pathogenesis of ALI predominantly entails the interplay of apoptosis and inflammation, ultimately culminating in cellular demise. These intricate processes encompass alterations within liver parenchymal cells and non-parenchymal cells, comprising hepatocytes and Kupffer cells. Pertinently, recent clinical, *in vivo*, and *in vitro* investigations have unveiled a compelling association between the activation of Kupffer cells, hepatocyte death, and the involvement of SIRT1, thereby shedding light on the underlying mechanisms driving liver injury.

The progression of ALI is contingent upon the inhibition of SIRT1 and its downstream signaling molecules within the hepatic milieu. *Kemelo et al. (2014)* demonstrated a

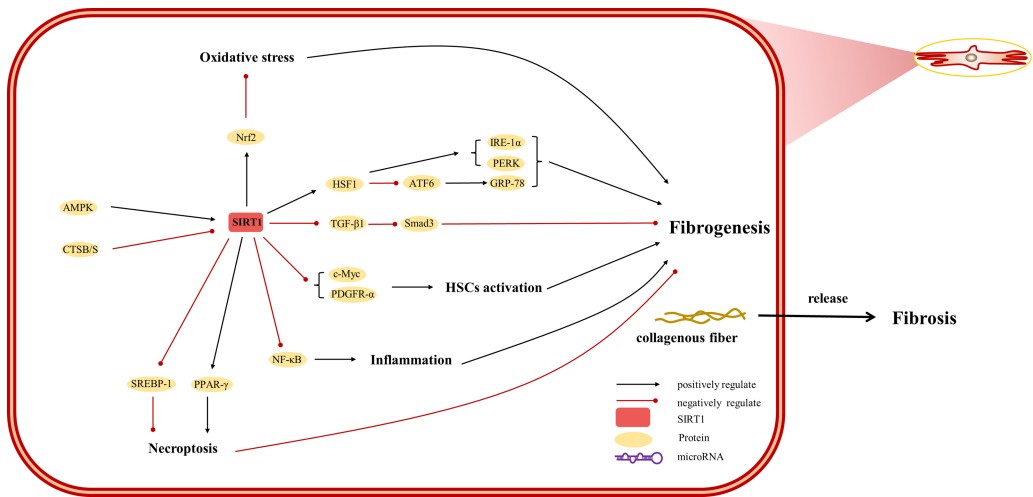

**Figure 4 Overview of the roles of SIRTs in human hepatic stellate cells.** (AMPK) adenosine 5′-monophosphate-activated protein kinase, (ATF6) activating transcription factor 6, (CTSB/S) cathepsin B/S, (GRP-78) 78-kD glucose-regulated protein, (HSF1) heat shock factor 1, (IRE-1$\alpha$) inositol-requiring protein 1$\alpha$, (NF-$\kappa$ B) nuclear factor-$\kappa$ B, (Nrf2) Nuclear factor erythroid 2-related factor 2, (PDGFR-$\alpha$) platelet-derived growth factor receptor alpha, (PERK) protein kinase R-like ER kinase, (PPAR) peroxisome proliferator-activated receptor, (Smad) small mothers against decapentaplegic protein, (SREBP-1) sterol regulatory element-binding protein-1.

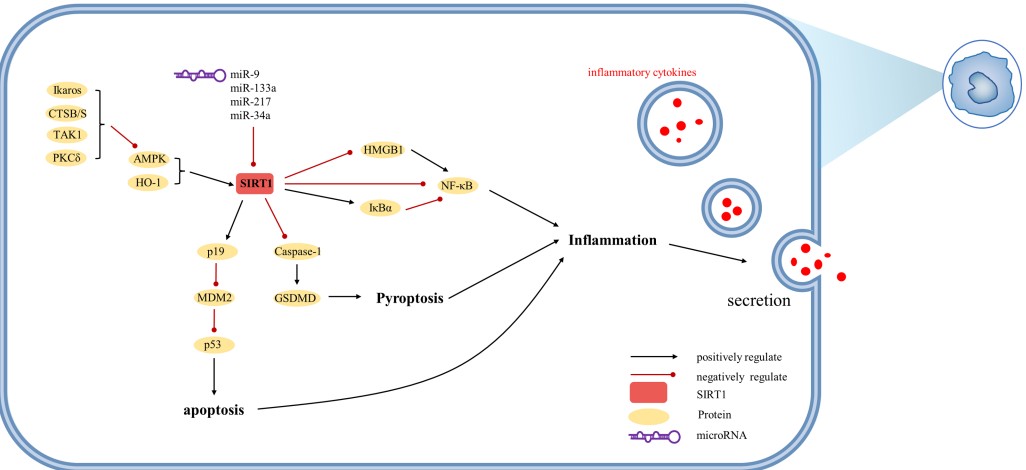

**Figure 5 Overview of the roles of SIRTs in Kupffer cells.** (AMPK) adenosine 5′-monophosphate-activated protein kinase, (CTSB/S) cathepsin B/S, (GSDMD) Gasdermin D, (HMGB1) high mobility group box 1, (HO-1) heme oxygenase-1, (I$\kappa$ B$\alpha$) inhibitor of kappa B alpha, (MDM2) mouse double minute 2, (PKC$\delta$) protein kinase C-delta, (TAK1) TGF-$\beta$1 activated kinase.

reduction in SIRT1 levels during lipopolysaccharide (LPS)-induced ALI, leading to impaired hepatic function. Subsequently, *Zhou et al. (2021)* corroborated the protective effects of SIRT1 in ALI models. While the majority of investigations have demonstrated the ameliorative impact of SIRT1 activation on liver injury, certain studies have reported

a contrasting perspective, suggesting that SIRT1 may contribute to disease progression and exacerbate tissue damage (*Cui et al., 2016*). Hence, elucidating the precise identities of upstream regulators and downstream effectors of SIRT1, as well as their respective roles, assumes paramount significance in furthering our understanding of ALI pathogenesis.

*The regulation of SIRT1 in liver tissue.* The hepatic tissue encompasses both liver parenchyma and mesenchyme, and exhibits noteworthy alterations in SIRT1 levels during injury. In the context of ALI, research has highlighted a pivotal interaction between microRNA-34a (miR-34a) and SIRT1. Elevated miR-34a levels in ALI models correlate with a suppression of SIRT1 expression (*El Shaffei et al., 2021*). This suppression is crucial as SIRT1 plays a protective role in hepatic tissues.

**Apoptosis and inflammatory**

The downregulation of SIRT1 initiates apoptosis and a pro-inflammatory cascade. This is evidenced by a rise in pro-apoptotic factors such as Bcl-2 associated X protein (Bax) and caspase-3, alongside a reduction in the anti-apoptotic protein Bcl-2, culminating in decreased liver cell survival (*Su et al., 2020*). Marked by the activation of nuclear factor-$\kappa$B (NF-$\kappa$B) and a subsequent increase in inflammatory cytokines like IL-1$\beta$, IL-6, and TNF-$\alpha$ (*Wang et al., 2018*; *Su et al., 2020*). Furthermore, NOD-like receptor pyrin domain containing 3 (NLRP3) inflammasome, an intricate complex crucial in the inflammatory response, exhibits diminished expression upon SIRT1 activation *via* the SIRT1 activator SRT1720, a process initiated by NF-$\kappa$B (*Khader et al., 2017*).

**Mitochondrial biogenesis and oxidative stress**

SIRT1 activation promotes the upregulation of peroxisome proliferator-activated receptor-gamma coactivator 1-alpha (PGC-1$\alpha$), which in turn boosts nuclear factor erythroid 2-related factor 2 (Nrf2) expression. This enhances mitochondrial biogenesis and antioxidant responses, evident in the induction of proteins like heme oxygenase-1 (HO-1) and GCLM in drug-induced liver injury scenarios (*Abdelzaher, Ali & El-Tahawy, 2020*; *El Shaffei et al., 2021*). Furthermore, the research by *El-Sheikh et al. (2023)* expands the protective roles of SIRT1 beyond drug-induced scenarios to include radiation-induced liver damage. In this context, upregulating SIRT1 leads to a suppression of poly ADP-ribose polymerase-1 (PARP-1) and forkhead box O1 (FoxO-1).

**Bile metabolism**

Activation of the SIRT1/HNF1 $\alpha$/FXR axis may represent a novel therapeutic target for ameliorating ALI and other herb-induced hepatotoxicity (*Liao et al., 2023*). This is in contrast to the findings of *Kemelo et al. (2017a)*, which demonstrated that SIRT1 negatively regulates HO-1, leading to a reduction in bilirubin production in an LPS-induced mouse model.

**Direct deacetylation**

Significantly, SIRT1 exerts direct deacetylation effects on pivotal transcription factors, including p53, NF-$\kappa$B, HMGB1, and FoxO1, as revealed by *Liu et al. (2017)*; *Fu et al. (2023)*. Intriguingly, the timing of feeding has emerged as a crucial determinant in the progression of ALI, due to its impact on modulating SIRT1 activity, as demonstrated in a mice model by *Oyama et al. (2014)*.

*The regulation of SIRT1 in hepatocytes.* The alterations observed in liver cells serve as the foundation for liver diseases, and the dysregulation of SIRT1 prominently contributes to these cellular changes. Notably, *Rada et al. (2018)* observed the inhibition of SIRT1 in both hepatocytes and macrophages within ALI models.

**Molecular interactions influencing SIRT1 activity**

Several molecules, including adenosine 5′-monophosphate-activated protein kinase (AMPK), are known to influence SIRT1, with a bidirectional relationship, as SIRT1 can also positively regulate AMPK (*Njeka Wojnarova et al., 2023*). AMPK enhances SIRT1 by increasing cellular NAD$^+$ levels (*Gao et al., 2020*), while triptolide has been observed to induce AMPK phosphorylation but suppress SIRT1 levels (*Yang et al., 2017*). MicroRNAs such as miR-19b, miR-128-3p, miR-485-3p, and miR-133a also modulate SIRT1 expression (*Liu et al., 2019a*; *Zhao et al., 2020*; *Chen et al., 2020a*; *Tang et al., 2023*). Moreover, *Yang et al. (2022a)* discovered that hsa_circ_0093884, a circular RNA derived from the NAD$^+$-dependent deacetylase, enhances SIRT1 levels through the depletion of ribosomal protein S3 (RPS3).

**Apoptosis and stress response**

SIRT1 exhibits diverse regulatory functions within hepatocytes. Notably, a recent investigation by *Tang et al. (2021)* demonstrated that the downregulation of SIRT1 exacerbates the upregulation of apoptosis-related proteins, namely caspase-3 and C/EBP-homologous protein (CHOP), in palmitic acid-induced liver injury. Inhibition of endoplasmic reticulum stress (ERS)-related proteins, including protein kinase R-like ER kinase (PERK), 78-kD glucose-regulated protein (GRP-78), and activating transcription factor 6 (ATF6), has also been reported as a consequence of SIRT1 activity (*Chen et al., 2020b*).

**Inflammation modulation**

By deacetylating NF-$\kappa$B, SIRT1 modulates its activity and the transcription of inflammatory genes. Additionally, SIRT1's interaction with HMGB1 is pivotal; the deacetylation of high mobility group box 1 (HMGB1) hinders its nuclear to cytoplasmic translocation, secretion, and overall expression, thereby attenuating its role in promoting inflammation (*Yu et al., 2019*). This dual mechanism of action, affecting both NF-$\kappa$B and HMGB1, leads to a significant reduction in pro-inflammatory cytokines such as IL-1$\beta$, IL-6, and TNF-$\alpha$ (*Chen et al., 2020a*).

**Bile metabolism**

SIRT1 also regulates bile metabolism by deacetylating farnesoid X receptor (FXR), which activates Nrf2 and diminishes the production of inflammatory cytokines (*Qu et al., 2018*; *Gao et al., 2020*). In *Sirt1* knockout hepatocytes, an enhanced transcriptional activity of p53 is observed, with selective induction of target genes like *Noxa* (*Cui et al., 2016*).

*The regulation of SIRT1 in macrophages.*

**MiRNA regulation** Liver inflammation is closely associated with Kupffer cells, the resident macrophages in the liver. In the context of cecal ligation and puncture (CLP)-induced liver injury, miR-133a has been shown to inhibit SIRT1 expression in macrophages (*Chen et al.,*

*2020a*). Similarly, miR-9, regulated by monocyte chemotactic protein 1 (MCP-1) induced protein (MCPIP1), has the ability to suppress SIRT1 (*Han et al., 2019*).

**Inflammation**

During the inflammatory response, SIRT1 exerts its inhibitory effect on NF-$\kappa$B signaling by activating inhibitor of kappa B alpha (I$\kappa$B $\alpha$), leading to reduced production of IL-1$\beta$, IL-6, and TNF-$\alpha$ in acetaminophen-induced RAW 264.7 cells (*Han et al., 2019*; *Rada et al., 2018*; *Chen et al., 2020a*). Notably, the release of IL-1$\beta$ by Kupffer cells directly impacts hepatocytes, resulting in damage (*Rada et al., 2018*).

**Oxidative stress**

SIRT1 in macrophages has been observed to increase the expression of manganese superoxide dismutase (MnSOD), enhance levels of glutathione (GSH) and p-c-Jun N-terminal kinase (JNK), promote IL-10 production, and decrease caspase-1 and nuclear-specific protein 1 (Sp1) in the context of liver injury (*Rada et al., 2018*). Collectively, these findings underscore the regulatory role of miRNAs in modulating SIRT1 activity and highlight the crucial involvement of SIRT1 in macrophage activation.

To summarize, recent studies showed that the main mechanisms of SIRT1 in ALI include the following points:

(1) SIRT1 is mainly regulated by AMPK and miRNAs in ALI.

(2) SIRT1 alleviates oxidative stress to improve liver injury mainly *via* Nrf2 in ALI.

(3) SIRT1 inhibits inflammation to reduce liver injury mainly *via* the regulation of NF-$\kappa$B in ALI.

(4) SIRT1 reduces hepatocyte apoptosis *via* p53 and caspases to play a hepatoprotective role in ALI.

### SIRT1 in ALD

ALD is a multifaceted condition that arises from prolonged and excessive alcohol intake, surpassing a defined daily threshold, and displays substantial interindividual variability (*Seitz et al., 2018*). Specifically, in this primer, excessive alcohol consumption is defined as the ingestion of more than 40 g of pure alcohol per day over an extended duration, thereby conferring the greatest risk for ALD (*Rehm et al., 2010*). The progression of ALD involves various pathological mechanisms, such as hepatic steatosis, oxidative stress, and inflammation.

The advancement of ALD encompasses changes occurring within the liver parenchyma and interstitium, affecting crucial cellular components such as hepatocytes, kupffer cells, and human hepatic stellate cells (HSCs). In addition to hepatocyte degeneration and necrosis, the activation of kupffer cells and HSCs represents a pivotal pathophysiological mechanism in the progression of ALD. Notably, SIRT1 has been established as an essential factor in the development of hepatocyte degeneration and necrosis. Consequently, targeting the activation of kupffer cells and HSCs holds promise as a potential therapeutic strategy to impede the progression of this disease.

*The regulation of SIRT1 in liver parenchymal components.* A notable decrease in SIRT1 levels in ALD models was observed (*Thompson et al., 2015*), highlighting the detrimental

effect of ethanol on SIRT1 expression. This was further elucidated by the finding that ethanol-induced upregulation of lipocalin-2 (LCN2) leads to SIRT1 downregulation (*Cai et al., 2016*), revealing specific molecular pathways affected by alcohol. Moreover, Fas-activated serine/threonine kinase (FASTK) was found to suppress SIRT1 expression by influencing mRNA stability *via* the binding protein human antigen R (HuR) (*Zhang et al., 2021*).

### $NAD^+$ metabolism and SIRT1 function in ALD

Parallel to these insights, the crucial role of $NAD^+$ metabolism in ALD and its impact on SIRT1 has been highlighted. The liver-specific silencing of the methylation-controlled J protein (MCJ) was shown to restore the $NAD^+$/NADH ratio, thereby enhancing SIRT1 function and mitigating lipogenesis while promoting lipid oxidation (*Goikoetxea-Usandizaga et al., 2023*). Additionally, the identification of miR-873-5p as a negative regulator of NAD metabolism and SIRT1 activity (*Rodríguez-Agudo et al., 2023*) underlines the intricate regulation by microRNAs. The importance of cellular energy balance in SIRT1 functionality was underscored by the discovery of AMP-activated protein kinase (AMPK) in activating SIRT1 in ALD (*Nagappan et al., 2020*).

### Protective mechanisms of SIRT1 in hepatocytes

The protective mechanisms of SIRT1 in ALD encompass a range of cellular processes. One such mechanism involves the enhancement of autophagy in hepatocytes, facilitated by SIRT1-mediated upregulation of RAB7 and lysosomal-associated membrane protein 2 (LAMP-2), as well as the suppression of mechanistic target of rapamycin (mTOR) signaling through induction of DEP domain-containing mTOR-interacting protein (DEPTOR) (*Shi et al., 2018*; *Chen et al., 2018*).

SIRT1 mitigates oxidative stress by promoting the expression of PGC-1 $\alpha$, glutathione peroxidase-2 (Gpx2), superoxide dismutase-1 (SOD1), and GSH levels (*Zhang et al., 2021*; *Fan et al., 2021*). Notably, intestinal deficiency of SIRT1 leads to altered liver iron concentrations, decreased ferroptosis, and elevated GSH levels (*Zhou et al., 2020*).

SIRT1 regulates the expression of splicing factor, arginine/serine-rich 10 (SFRS10), which in turn reduces lipin-1 $\beta$ expression, resulting in the attenuation of hepatic steatosis and resistance against liver fibrosis, as evidenced by reduced collagen1 $\alpha$, tissue inhibitor of metalloproteinase 1 (TIMP-1), and $\alpha$-smooth muscle actin ($\alpha$-SMA) levels (*Yin et al., 2014*). Lastly, SIRT1 exerts anti-inflammatory effects by downregulating nuclear factor of activated T cells c4 (NFATc4) and MCP-1 which was confirmed in the livers of *Sirt1* knockout mice (*Yin et al., 2014*; *Zhang et al., 2021*).

*The regulation of SIRT1 in liver non-parenchymal cells.* In the progression of ALD, non-parenchymal cells such as HSCs and Kupffer cells play a critical role, particularly in the development of alcoholic steatohepatitis. Within the liver microenvironment, these cells significantly contribute to inflammation, fibrosis, and dysregulation of lipid metabolism, key hallmarks of ALD.

**HSCs**

In HSCs, the regulation of SIRT1 is intricately influenced by AMPK and liver kinase B1 (LKB1), with AMPK acting as an energy sensor that activates SIRT1, affecting pathways related to inflammation and fibrosis (*Yang et al., 2016*; *Bai et al., 2016b*). This activation leads SIRT1 to reduce inflammation by deacetylating p65 NF-κB and to regulate lipid metabolism, balancing sterol regulatory element-binding protein-1 (SREBP-1) and peroxisome proliferator-activated receptor gamma (PPAR-γ) activities. Furthermore, the age-related decline in SIRT1, correlated with increased hepatic fibrosis markers, emphasizes its critical role in liver aging and fibrosis (*Ramirez et al., 2017*).

**Macrophages**

In Kupffer cells, SIRT1 is inhibited by miR-217, impacting lipid metabolism *via* the AMPK/SIRT1 pathway (*Yin et al., 2015*), illustrating the complex regulation of SIRT1 in liver immune cells. Additionally, silencing miR-34a increases Sirt1 expression in macrophages, essential for managing steatohepatitis and angiogenesis during alcohol-induced liver injury, indicating a regulatory interplay between miR-34a and SIRT1 in macrophages and highlighting their role in ALD pathology (*Wan et al., 2023*).

In summary, SIRT1 reduces lipid accumulation by enhancing autophagy, inhibiting lipid production, and promoting lipid metabolism in hepatocytes or HSCs to obstruct the occurrence of fatty liver. Besides, SIRT1 alleviates liver injury by inhibiting inflammation and hepatocyte apoptosis in ALD. Importantly, the regulation of SIRT1 in ALD is significantly influenced by NAD+ metabolism.

### SIRT1 in NAFLD

Dietary and lifestyle factors play pivotal roles in the development of NAFLD, which is characterized by the presence of hepatic steatosis as confirmed by imaging or histological examination, with other liver diseases adequately excluded (*Chen et al., 2006*). NAFLD encompasses two distinct clinical entities: nonalcoholic fatty liver and nonalcoholic steatohepatitis, the latter requiring liver biopsy for definitive diagnosis (*Tokushige et al., 2021*).

*Upstream regulation of SIRT1.* Hepatic SIRT1 levels serve as a reflection of molecular changes associated with NAFLD. The regulation of SIRT1 in NAFLD involves multiple pathways. Notably, liver PARP levels, which can result in depletion of ATP and $NAD^+$ and subsequent inhibition of SIRT1, progressively increase during the development of NAFLD (*Gariani et al., 2017*). In liver hepatocytes, SIRT1 is predominantly regulated by AMPK and miRNAs (*Han et al., 2023*). MiR-146B, miR-29a, and miR-22 have been identified as inhibitors of SIRT1 (*Sui et al., 2021*; *Yang et al., 2020b*; *Azar et al., 2020*). MiR-29a inhibits SIRT1 by inactivating glycogen synthase kinase 3 beta (GSK3β), while the effect of miR-22 is modulated by the cannabinoid-1 receptor (CB1R)/p53 pathway. Furthermore, the overexpression of serine/threonine protein phosphatase 2A (PP2A) in NAFLD reduces SIRT1 through dephosphorylation of AMPK (*Chen et al., 2019*). AdipoR1, an upstream molecule of LKB1, promotes SIRT1 *via* AMPK by increasing $NAD^+$ levels, while SIRT1

does not appear to directly affect AMPK (*Li et al., 2022b*). Additionally, consistent with previous reports, SIRT1 is regulated by PARP1 in NAFLD (*Ye et al., 2019*).

*SRIT1 in lipid metabolism and production.* Lipid accumulation represents a critical stage in the development of NAFLD. While a study by *Li et al. (2022b)* suggested that SIRT1 does not influence AMPK, another investigation reported that SIRT1 directly deacetylates LKB1, thereby promoting AMPK activity and contributing to lipid homeostasis maintenance (*Jia et al., 2016*). Furthermore, SIRT1 has been shown to enhance hepatic insulin sensitivity and regulate lipid homeostasis by promoting adipose triglyceride lipase (ATGL) expression and deacetylating transcription factors FoxO1 and PGC-1 $\alpha$ (*Jia et al., 2016*; *Sui et al., 2021*; *Chen et al., 2019*). In hepatocytes, SIRT1 promotes the expression of key regulators such as carnitine palmitoyltransferase-1A (CPT1A), mitochondrial transcription factor A (TFAM), and PPAR $\alpha$ to modulate hepatic lipid metabolism (*Li et al., 2022b*; *Yang et al., 2020b*; *Azar et al., 2020*).

*SRIT1 in other processes contributing NAFLD.*
**Mechanisms of SIRT1 regulation and impact on inflammatory pathways.** Inflammation plays a pivotal role in the clinical classification of NAFLD and is indicative, to some extent, of disease progression and outcome. SIRT1, a key regulator in NAFLD, exerts beneficial effects by mitigating liver inflammation and oxidative stress. This is achieved through the inhibition of NLRP3 (*Chen et al., 2020b*). The impact of SIRT1 on the molecular aspects of NAFLD has also been demonstrated in cell-based studies. Specifically, SIRT1 attenuates inflammation by reducing the expression and secretion of HMGB1 (*Zeng et al., 2015*). In macrophages, SIRT1 inhibits the activity of p65, thereby diminishing inflammation in NAFLD (*Niu et al., 2018*). Notably, the reduction of SIRT1 levels by cathepsin B/S (CTSB/S) proteolytic cleavage impedes the deacetylation of p65, leading to the induction of MCP-1. This process has been observed in hepatocytes, HSCs, and Kupffer cells (*De Mingo et al., 2016*).
**SIRT1-mediated autophagy in hepatocytes of NAFLD models**
The hepatocytes of NAFLD models have provided further evidence of the diverse effects mediated by SIRT1. In hepatocytes, SIRT1 stimulates autophagy by upregulating autophagy-related gene 7 (ATG7) and increasing the levels of microtubule-associated protein 1A/1B-light chain 3 (LC3-II) (*Hong et al., 2018*). This augmentation of autophagy in liver cells is substantiated by the observed elevation of LC3-II levels and concurrent reduction of p62 (*Ren et al., 2019*).
**SIRT1's involvement in oxidative stress and tumor-suppressive role**
SIRT1 mitigates oxidative stress in hepatocytes by inhibiting the generation of reactive oxygen species (ROS) and promoting the expression of Nrf2, SIRT3, SOD, and glutathione peroxidase (GSH-Px) (*Li et al., 2022b*). Additionally, SIRT1 appears to exert a tumor-suppressive role in high-fat choline-deficient (HFCD) diet-induced mice by modulating the FXR and regulating the suppressor of cytokine signaling 3 (SOCS3)/Janus kinase 2 (Jak2)/signal transducer and activator of transcription 3 (STAT3) signaling axis (*Attia et al., 2021*).

### SIRT1 in IRI

The hepatic tissue demonstrates a high susceptibility to hypoxic conditions, particularly in the context of liver transplantation, where the occurrence of IRI poses significant risks. IRI manifests in the liver and encompasses a cascade of interconnected processes, including autophagy, oxidative stress, and metabolic dysfunction, affecting various cellular populations. The intricate cellular and molecular events underlying liver IRI can be linked to key clinical factors associated with IRI in the context of liver transplantation, such as donor organ steatosis, duration of ischemia, donor age, and coagulation abnormalities in both the donor and recipient (*Dar et al., 2019*).

Oxidative stress, apoptosis, and inflammation represent pivotal mechanisms underlying liver ischemic injury. In this context, the involvement of SIRT1 in modulating these processes has garnered significant attention. Experimental evidence obtained from murine models has substantiated the inhibitory effect of interferon regulatory factor (IRF) 9 and sorbitol dehydrogenase (SDH) on SIRT1 during IRI (*Wang et al., 2015*; *Zhang, Li & Liu, 2015a*). The regulatory influence of SDH on SIRT1 is achieved through its modulation of $NAD^+$ levels, thereby affecting SIRT1 activity and function.

*SIRT1 alleviates oxidative injury and apoptosis in IRI.* Hepatic ischemia triggers oxidative stress, leading to hepatocellular damage and apoptosis.

**Oxidative stress**

SIRT1 enhances endothelial nitric oxide synthase (eNOS) activity producing produces nitric oxide (NO), which attenuate oxidative stress by scavenging superoxide anions and other reactive oxygen species (ROS), thus protecting hepatocytes from ischemia-induced damage. On the other hand, SIRT1 activates pAMPK, a key energy sensor in cells. Activation of AMPK modulates metabolic pathways to reduce oxidative stress, such as inhibiting the production of ROS from mitochondria by adjusting metabolic flux. Further, SIRT1 inhibits the adaptor protein p66shc to prevents excessive ROS production, particularly in the mitochondria, thereby protecting hepatocytes from oxidative stress-induced apoptosis (*Yan et al., 2014*; *Pantazi et al., 2014*).

**Apoptosis**

Perturbations in eNOS, pAMPK, and p66shc have been implicated in apoptosis observed in IRI models. The diminished expression of SIRT1 during IRI results in the upregulation of p53, cleaved caspase3/9, p-p38, and downregulation of extracellular signal-regulated kinase (ERK) and heat shock protein 70 (HSP70), culminating in hepatocellular death (*Khader et al., 2016*; *Pantazi et al., 2015b*). Simultaneously, SIRT1 in hepatocytes is responsible for regulating anti-apoptotic proteins such as Bcl-2 and X-linked inhibitor of apoptosis protein (XIAP), and it mitigates apoptosis triggered by cold stress. This regulation by SIRT1 helps in reducing the activation of gasdermin E (GSDME) and the release of interleukin-18 (IL18), further protecting the hepatocytes from death (*Kadono et al., 2023*).

*Inflammation is inhibited by SIRT1 in IRI.*

**Molecular changes of hepatocytes** Ischemia-reperfusion injury induces hepatocellular inflammation as a pathological process. Hepatic ischemia triggers a reduction in SIRT1

levels concurrent with the upregulation of HMGB1 in isolated liver models (*Zaouali et al., 2017*). SIRT1 has demonstrated its anti-inflammatory effects by inhibiting macrophage inhibitory protein (MIP)-2, NF-κB, and p53 through the activation of FXR signaling pathway (*Khader et al., 2016*; *Sun et al., 2020*). Additionally, SIRT1-mediated upregulation of miR-182 suppresses the NLRP3 inflammasome through X-box binding protein 1 (XBP1) inhibition, thereby alleviating inflammation (*Li et al., 2021*).

**Molecular changes of immune cell**

SIRT1 exerts inhibitory effects on liver macrophage activation. In macrophages, SIRT1 can be upregulated by HO-1 to modulate autophagy processes (*Nakamura et al., 2018*; *Nakamura et al., 2017*). Similar to hepatocytes, SIRT1 activation is facilitated by AMPK, with recent evidence indicating that Ikaros restricts AMPK activity in this context (*Kadono et al., 2022*). Inflammatory factors and p53 play pivotal roles in the regulatory functions of SIRT1 in macrophages. Studies have demonstrated that p53 directly inhibits SIRT1 or acts indirectly *via* miR-34a, while SIRT1 reduces p53 acetylation (*Kim et al., 2015*). However, in contrast to hepatocytes, SIRT1-induced alternative reading frame (Arf; p19 in mouse; p14 in human) inhibits mouse double minute 2 (MDM2), thereby preventing ubiquitin-mediated degradation of p53. The p19 axis involving HO1/SIRT1/p19 upregulates p53 tumor suppressor protein, which attenuates macrophage activation (*Nakamura et al., 2017*). Furthermore, SIRT1 mitigates immune cell recruitment by reducing the production of inflammatory mediators such as IL-1β and IL-18 through the inhibition of proteolytic cleavage of Gasdermin D (GSDMD) (*Kadono et al., 2022*). Notably, SIRT1-mediated deacetylation of HMGB1 plays a critical role in the expression of IL-1β and IL-18, as well as the active secretion of HMGB1 by macrophages (*Sun et al., 2017*).

### *SIRT1 in cholestatic injury*

In the clinical setting, the American College of Gastroenterology (ACG) characterizes cholestatic injury as a notable increase in alkaline phosphatase levels relative to AST and ALT levels (*Kwo, Cohen & Lim, 2017*). Bilirubin, predominantly existing in an unconjugated form, becomes a relevant indicator of hepatocellular disease or cholestasis when the levels of conjugated bilirubin are elevated. It is noteworthy that cholestatic injury has the potential to advance to cirrhosis and even hepatocellular carcinoma.

*SIRT1 reduce bile production via FXR.* In the hepatic parenchyma, SIRT1 plays a crucial role in the regulation of bile metabolism, primarily through its interaction with the FXR. It not only directly deacetylates FXR, thereby mitigating cholestasis and hepatotoxicity, but also enhances FXR activity *via* the modulation of hepatic nuclear receptor-1a (HNF-1a) (*Zhu et al., 2018b*). This activation of FXR by SIRT1 influences various aspects of BA management, including uptake, excretion, synthesis, and metabolism. Notably, the SIRT1-FXR interaction leads to the repression of BA synthesis by inducing the small heterodimer partner (SHP) to interact with liver receptor homolog-1 (LRH-1) (*Blokker et al., 2019*). Moreover, FXR plays a critical role in regulating BA transporters such as the Na+-dependent taurocholate cotransport peptide (NTCP) and the bile salt export pump (BSEP), as well as multidrug resistance-associated proteins (MRPs), effectively reducing the

harmful effects of excessive BA accumulation. Additionally, SIRT1 boosts BA metabolism by upregulating key metabolic enzymes through FXR signaling (*Yang et al., 2020a*).

*SIRT1 inhibits inflammation in cholestatic injury.* Clinical and experimental investigations have consistently demonstrated the anti-inflammatory properties of SIRT1 in the context of cholestatic liver injury, as evidenced by the reduction in key inflammatory mediators including NF-$\kappa$B, transforming growth factor$\beta$1 (TGF-$\beta$1), TNF-$\alpha$, and IL-6 (*Zhao et al., 2019*; *Kabil, 2018*). Notably, in Kupffer cells, SIRT1 exerts its anti-inflammatory effects by suppressing the production of pro-inflammatory cytokines such as interferon-$\gamma$ (IFN-$\gamma$). However, this regulatory mechanism is hampered in the presence of poly I:C-induced upregulation of TGF-$\beta$1 activated kinase (TAK1) (*Li et al., 2020b*). It is important to note that the scenario changes in models of bile duct ligation/LPS-induced liver injury, where SIRT1 promotes heightened liver inflammation and liver injury through the activation of mTORC1 (*Isaacs-Ten et al., 2022*).

*SIRT1 prevents the progression to liver fibrosis.* Cholestasis represents a significant etiological factor in the development of liver fibrosis. To impede the progression of cholestasis towards liver fibrosis, SIRT1 exerts inhibitory effects on HSCs. Notably, SIRT1 modulates the acetylation status of heat shock factor 1 (HSF1), thereby extending the binding duration of HSF1 to the promoter region of heat shock genes. This, in turn, mitigates ERS, as evidenced by the reduction in PERK, inositol-requiring protein 1 $\alpha$ (IRE-1 $\alpha$), ATF6, and GRP78 (*Zhu et al., 2018a*). ERS instigates the activation of fibrogenic genes and contributes to the progression of liver fibrosis in HSCs. Additionally, research findings have demonstrated that the upregulation of SIRT1 attenuates the expression of $\alpha$-SMA and impedes the accumulation of collagen, further underscoring its role in preventing liver fibrosis (*Kabil, 2018*).

*Others.* Noteworthy variations in molecules shape the regulation of SIRT1. Specifically, miR-34a, an inhibitor of SIRT1, is promoted by JNK through p53 activation (*Kulkarni et al., 2016*). Furthermore, hepatocytes overexpressing nerve growth factor (NGF) exhibit simultaneous enhancement of SIRT1 and phosphorylation of NF-$\kappa$B (*Tsai et al., 2018*).

Inhibition of SIRT1 leads to the activation and upregulation of p53 in both *in vivo* and *in vitro* models of cholestasis, underscoring the interplay between these molecules (*Zhao et al., 2019*). Additionally, SIRT1's activation of AMPK results in increased levels of Nrf2, which boosts the expression of antioxidant response genes such as HO-1 and elevates levels of antioxidants like glutathione (GSH) and superoxide dismutase (SOD) (*Li et al., 2022a*; *Chen et al., 2023*).

### SIRT1 in liver fibrosis

Liver fibrogenesis is a complex and orchestrated molecular, cellular, and tissue process that underlies the pathological buildup of extracellular matrix (ECM) components, leading to the development of liver fibrosis. This process is sustained by a diverse population of hepatic myofibroblasts (MFs) (*Parola & Pinzani, 2019*). The progression of liver fibrosis

not only results in the destruction of the hepatic lobule structure but also disrupts normal liver function, thereby compromising overall hepatic homeostasis.

*SIRT1 reduces the activated HSCs.* The activation of HSCs is a central event in the development of liver fibrosis and SIRT1 is a key regulator of the activation of HSCs. A recent study suggests that defects in SIRT1 that are age-dependent can lead to the activation of HSCs (*Adjei-Mosi et al., 2023*).

**Modulating TGF-$\beta$1 signaling**

The action of SIRT1 leads to the suppression of TGF-$\beta$1 signaling, which results in a reduction in the expression of fibrogenic markers such as collagen I and $\alpha$-SMA. This mechanism is key to understanding how SIRT1 exerts anti-fibrotic effects in the liver (*Ma et al., 2019*).

**Combating oxidative stress**

Oxidative stress is a significant factor that contributes to the activation of HSCs. SIRT1 enhances the antioxidant response by upregulating Nrf2. This leads to increased production of enzymes like HO-1 and glutathione-S-transferase (GST), reducing oxidative stress in HSCs (*Gu et al., 2016*).

**SIRT1's interaction with the AMPK pathway in HSCs**

SIRT1 interacts with the AMPK pathway, playing a role in HSC activation under oxidative stress. The influence of SIRT1 in this pathway, particularly in the context of SOD3 depletion's impact on HSC activation, was highlighted (*Sun et al., 2021*).

**Induction of necroptosis in HSCs**

SIRT1 is involved in inducing the degradation of the Notch intracellular domain (NICD) and activating RIP1/3, leading to necroptosis in HSCs. This pathway represents an additional mechanism by which SIRT1 can reduce the activation of HSCs (*Sun et al., 2022*).

*SIRT1 protect hepatocytes and inhibit fibrosis.*

**MiR-34a/SIRT1/p53 signaling pathway** P53 induces the expression of miR-34a, which in turn inhibits SIRT1, leading to further activation of p53. The inhibition of SIRT1 by miR-34a could disrupt cellular homeostasis, promoting processes like cell death and inflammation that contribute to fibrosis. This feedback loop is active in hepatocytes, particularly in CCl4-induced liver fibrosis, but not in HSCs (*Tian et al., 2016*).

**FGF21 and SIRT1**

SIRT1 induction by FGF21 enhances autophagy and reduces inflammation, which helps protect against CCl4-induced liver injury (*Yang et al., 2022b*). This suggests that the presence and activation of SIRT1 are crucial for mitigating liver injury and, by extension, preventing fibrosis.

**Metabolic processes**

The inhibition of SIRT1, as part of the AMPK/SIRT1/FXR pathway, impairs the liver's ability to regulate metabolic processes and stress responses, potentially contributing to the development of fibrosis (*Keshk et al., 2019*). Similarly, aging leads to the downregulation of SIRT1 in hepatocytes, and this reduction in SIRT1 activity in aging livers may further

impair the cells' capacity to respond to stress and maintain metabolic balance, thereby exacerbating the risk of fibrotic changes (*Dai et al., 2023*)

*SIRT1 in macrophages.* Activating SIRT1 in macrophages leads to the down-regulation of the NLRP3 pathway. The down-regulation of NLRP3 and subsequent changes in NF-$\kappa$B activity indirectly inhibit the activation of HSCs. This cascade of events helps alleviate hepatic inflammation and fibrosis (*He et al., 2023*). Specifically, in Kupffer cells, Protein Kinase C-delta (PKC $\delta$) inversely regulates SIRT1, leading to the activation of the transcription factor p65. The involvement of PKC $\delta$ suggests its role in inflammatory fibrosis through a SIRT1-dependent mechanism (*Lee et al., 2019*).

*Contradictory roles of SIRT1.* In a study involving CCl4-treated mice, an upregulation of hepatic SIRT1 was observed to exacerbate liver injury (*Kemelo et al., 2017b*). This contradicts the typically protective role ascribed to SIRT1, suggesting a more nuanced function in liver diseases, indicating that SIRT1's role in liver health is not straightforwardly beneficial; rather, its impact may depend on specific contexts or levels of expression.

**Preliminary summary**

Collectively, SIRT1 exhibits multifaceted roles in liver diseases encompassing inflammation, autophagy, oxidative stress, apoptosis, lipid metabolism, bile production, and fibrogenesis in direct or indirect ways. These roles are modulated both directly and indirectly.

Key regulators of SIRT1 include AMPK, which promotes SIRT1 activity, and miRs that act as inhibitors. The interplay between AMPK, miRs, and SIRT1 is crucial in understanding SIRT1's role in liver health.

SIRT1 plays a distinctive role in liver diseases caused by different factors. In the process of liver disease, the level of SIRT1 in three kinds of cells will change, namely hepatocytes, HSCs, and Kupffer cells. Particularly, SIRT1 is repressed in hepatocytes and Kupffer cells in ALI and IRI, which leads to inflammation, oxidative stress, and apoptosis. The changes of SIRT1 in either parenchyma, which is related to autophagy and oxidative stress, or mesenchyma, which affects inflammation and fibrogenesis participate in the progress of ALD together. The effect of SIRT1 on NAFLD is mainly reflected in its regulation of lipid metabolism in hepatocytes, and it can also affect NAFLD through other pathways. In cholestatic injury, SIRT1 reduce bile production *via* FXR and resist inflammation and fibrosis. SIRT1 also reduces hepatocyte apoptosis and inhibits the activation of HSCs and Kupffer cells to suppress fibrosis.

Interestingly, the regulation of SIRT1 in liver diseases appears highly adaptive. For example, it can upregulate HO-1 in response to cisplatin-induced liver injury to counteract oxidative stress, but decrease HO-1 in D-galactosamine/LPS-induced models to reduce bilirubin production. SIRT1 activation has been shown to reduce hepatocyte apoptosis but promote necrosis in HSCs. Besides, AMPK is regarded as a upstream regulator of SIRT1, as SIRT1 can positively affect it in some conditions but not in other cases of ALI models.

## Therapy of targeting SIRT1

The expanding recognition of SIRT1's significance in liver injury has spurred the discovery of numerous drugs targeting this enzyme. These pharmacological agents exhibit diverse levels of selectivity and can be categorized into different classes. In this section, our objective is to provide a comprehensive summary of recent investigations focusing on drugs that specifically target SIRT1 in the context of liver injury. By these studies, we aim to elucidate the therapeutic potential and underlying mechanisms of SIRT1, thereby contributing to a deeper understanding of their role in liver pathophysiology. Related drugs and mechanism are summarized in Table 1.

### Drug repurposing

In recent years, increasing literatures have emerged highlighting the therapeutic potential of drugs targeting SIRT1 which originally used to treat other diseases. Notably, these drugs have shown promise in addressing liver injury stemming from ALI and IRI, as well as in the context of ALD, NAFLD, and cholestatic injury. The increasing prevalence of such cases underscores the importance of exploring the role of SIRT1 modulation as a potential therapeutic strategy for a wide range of liver diseases.

Nicotinic acid, a derivative of vitamin B3 that has been utilized in clinical therapy for several decades, has demonstrated efficacy in mitigating acetaminophen-induced ALI through activation of the SIRT1/Nrf2 antioxidative pathway (*Hu et al., 2021*). Ketotifen, a mast cell stabilizer commonly prescribed for asthma treatment, has also exhibited similar hepatoprotective effects (*Abdelzaher, Ali & El-Tahawy, 2020*). Alogliptin, primarily used as an antidiabetic agent, has been shown to modulate the SIRT1/FoxO1 axis to ameliorate hepatic injury induced by cyclophosphamide in a mouse model (*Salama et al., 2020*). N-acetylcysteine (NAC), a long-standing therapeutic agent with a history spanning nearly six decades, has recently been found to enhance SIRT1 expression, leading to anti-apoptotic effects mediated by caspase-3 and Bax in models of ALI (*Harchegani et al., 2022*). Moreover, emerging drugs such as N-Dimethylglycine (DMG) have also shown associations with SIRT1 in the context of ALI (*Bai et al., 2016a*). These findings underscore the potential of targeting SIRT1 and its related pathways as a promising avenue for therapeutic intervention in liver injury.

Similar observations have been documented in various liver injury models. Melatonin, a commonly used health product, has been shown to possess anti-inflammatory, anti-oxidative stress, and enhanced autophagy properties in mice with NAFLD (*Bonomini et al., 2018*; *Ren et al., 2019*). S-adenosylmethionine has also been identified as a protective agent against ethanol-induced oxidative stress, and its mechanism of action involves modulation of SIRT1 (*Stiuso et al., 2016*). Cilostazol, a widely utilized vasodilating antiplatelet drug for the treatment of thrombotic vascular disease, has demonstrated a dose-dependent hepatoprotective effect in a murine model of bile duct ligation-induced liver injury through various mechanisms, including upregulation of SIRT1 (*Kabil, 2018*). These findings underscore the potential therapeutic role of targeting SIRT1 in mitigating liver injury across different pathological conditions.

**Table 1  Medicines targeting SIRT1 in liver injury.**

| Medicines | Mechanisms | Refs |
|---|---|---|
| Alogliptin | SIRT1/FoxO1 | *Salama et al. (2020)* |
| Ampelopsin | Autophagy | *Ma et al. (2019)* |
| Andrographolide | Unclear | *Wang et al. (2019)* |
| Atractylenolide III | AMPK/SIRT1 | *Li et al. (2022b)* |
| Astragaloside IV | AMPK/SIRT1 | *Qin et al. (2023)* |
| Baicalin | SIRT1/FXR | *Yang et al. (2020a)* |
| Berberine | microRNA-146b/SIRT1/FoxO1; AMPK/SIRT1 | *Sui et al. (2021)*; *Zhu et al. (2023)* |
| Betanin | SIRT1/PGC-1$\alpha$/Nrf2 | *El Shaffei et al. (2021)* |
| Botulin | AMPK/SIRT1 | *Bai et al. (2016b)* |
| Butein | SIRT1/NF-$\kappa$B | *Ghare et al. (2023)* |
| Celastrol | SIRT1/FXR | *Zhao et al. (2019)* |
| Chlorogenic acid | SIRT1/FXR | *Zhu et al. (2018b)* |
| Cichorium intybus linn | SIRT1/FXR | *Keshk et al. (2019)* |
| Cilostazol | Unclear | *Kabil (2018)* |
| Corynoline | SIRT1/Nrf2 | *Sun et al. (2023)* |
| Cryptotanshinone | AMPK/SIRT1 | *Nagappan et al. (2020)* |
| Cudratricusxanthone A | SIRT1/NF-$\kappa$B | *Lee et al. (2018)* |
| Curcumol | ERS | *Sun et al. (2022)* |
| Curcuma Aromatica Salisb | SIRT1/ HO-1 | *Kim et al. (2023)* |
| Demethyleneberberine | microRNA-146b/SIRT1, AMPK/SIRT1 | *Zhang et al. (2015b)* |
| Dioscin | SIRT1/Nrf2 | *Gu et al. (2016)* |
| Ginkgetin | SIRT1/ FOXO1/NF-$\kappa$B | *Alherz et al. (2023)* |
| Ginsenoside Rg1 | SIRT1/AMPK/Nrf2 | *Gao et al. (2023)* |
| Hesperidin | NLRP3/ SIRT1/FoxO | *Abo El-Magd et al. (2023)* |
| Imperatorin | AMPK/SIRT1/FXR | *Gao et al. (2020)* |
| Korean red | AMPK/SIRT1 | *Han et al. (2015)* |
| Kaempferol | SIRT1/AMPK | *Li et al. (2023)* |
| Losartan | Unclear | *Pantazi et al. (2015a)* |
| Lycium barbarum Polysaccharide | AMPK/SIRT1 | *Jia et al. (2016)* |
| Myricetin | Autophagy | *Rostami, Baluchnejadmojarad & Roghani (2023)* |
| Nicotinic acid | SIRT1/Nrf2 | *Hu et al. (2021)* |
| Pachymic Acid | SIRT1/HMGB1 | *Xue et al. (2023)* |
| Pioglitazone | SIRT1/Nrf2/HO-1 | *Kamel & Elariny (2023)* |
| Pterostilbene | SIRT1/FOXO1, SIRT1/NF-$\kappa$B, SIRT1/p53 | *Liu et al. (2017)* |
| Quercetin | Apoptosis; SIRT1/PGC-1$\alpha$; SIRT1/NF-$\kappa$B; SIRT1/Nrf2 | *Kemelo et al. (2016)*; *Zhang et al. (2019)*; *Çomaklı et al. (2023)*; *Abd El-Emam et al. (2023)* |

Wang et al. (2024), *PeerJ*, DOI 10.7717/peerj.17094

**Table 1** (*continued*)

| Medicines | Mechanisms | Refs |
|---|---|---|
| Rebamipide | Unclear | *Gendy, Abdallah & El-Abhar (2017)* |
| Resveratrol | SIRT1/HMGB1, SIRT1/NF-$\kappa$B/p53 | *Xu et al. (2014)*; *Yu et al. (2019)* |
| Rosa Rugosa Extract | AMPK/SIRT1 | *Lei et al. (2023)* |
| S-adenosylmethionine | Oxidative stress | *Stiuso et al. (2016)* |
| Salidroside | AMPK/SIRT1 | *Xu et al. (2023)*; *Gao et al. (2023)* |
| Salvianolic acid A | AMPK/SIRT1/HSF1, autophagy | *Li et al. (2020a)*; *Zhu et al. (2018a)* |
| Salvianolic acid B | SIRT1/PGC-1$\alpha$, SIRT1/HMGB1 | *Su et al. (2020)*; *Zeng et al. (2015)* |
| Sesquiterpenoids | SIRT1/Nrf2, SIRT1/NF-$\kappa$B | *Wang et al. (2018)* |
| Silymarin | Unclear | *Chang et al. (2019)* |
| Sinensetin | SIRT1/Nrf2 | *Lin et al. (2023)* |

## Treatment for specific liver diseases

Numerous plant-derived compounds have demonstrated significant regulatory effects on SIRT1, with traditional Chinese medicine (TCMs) emerging as a prominent source of these bioactive components. The utilization of TCMs has garnered considerable interest in the field, primarily attributed to their ability to modulate SIRT1 and confer hepatoprotective effects. However, it is worth noting that the exploration of targeted drugs specifically aimed at SIRT1 in the context of IRI remains limited. Future research endeavors are warranted to investigate and identify novel compounds with the potential to target SIRT1 in the context of IRI.

### Drugs for ALI.

**Antioxidative stress modulators in SIRT1 targeted therapies**

**Resveratrol, quercetin, betanin, pterostilbene:** These compounds are recognized for their antioxidative properties. They activate the SIRT1/Nrf2 pathway, a crucial axis in cellular defense against oxidative stress. By upregulating this pathway, these compounds enhance the antioxidative capacity of liver cells, thereby reducing oxidative damage.

**Ginkgetin, curcuma aromatica salisb:** These compounds represent another class with strong antioxidative effects. These agents modulate the SIRT1/FOXO-1/NF-$\kappa$B signaling pathway. The modulation of this pathway is crucial for controlling oxidative stress levels in hepatocytes.

**Anti-inflammatory agents in SIRT1 targeted therapies**

**Quercetin, sinensetin, hesperidin:** These compounds upregulate SIRT1, leading to the modulation of key inflammatory markers such as NF-$\kappa$B, TNF $\alpha$, and IL-6. SIRT1 activation results in the suppression of these pro-inflammatory cytokines, thereby reducing the inflammatory response in liver tissues.

**Corynoline, astragaloside IV:** Both Corynoline and Astragaloside IV target the SIRT1/Nrf2 signaling pathway. The Nrf2 pathway is known for its role in oxidative stress response, but its interplay with SIRT1 also significantly impacts inflammation. By modulating this pathway, these compounds reduce the inflammatory processes in the liver.

### Enhancers of metabolic regulation in SIRT1 targeted therapies

**Salvianolic acid B, imperatorin:** These compounds influence critical metabolic pathways, notably the AMPK/SIRT1/FXR cascade. AMPK and SIRT1 synergistically improve metabolic efficiency, while the FXR plays a pivotal role in bile acid metabolism, fatty acid synthesis, and glucose homeostasis.

**Ethanol extract of rosa rugosa:** The ethanol extract activates the LKB1/AMPK/Nrf2 cascade, subsequently upregulating Sirt1. LKB1 acts upstream of AMPK, fostering its activation, which in turn enhances the antioxidative and metabolic regulatory functions of Nrf2 and SIRT1.

### Modulators of autophagy and apoptosis in SIRT1 targeted therapies

**Salidroside, myricetin:** Both compounds modulate the AMPK/SIRT1 signaling pathway, a key regulator of cellular energy metabolism. Through this modulation, they influence autophagic and apoptotic pathways in liver cells.

**Pterostilbene, quercetin:** These compounds have the unique ability to directly deacetylate key transcription factors involved in cell survival and apoptosis. This action impacts various cellular pathways and gene expressions critical for liver cell health (all the cites of this part will be found in Table 1).

*Drugs for ALD.*

### Modulators of AMPK/SIRT1 pathway

Demethyleneberberine affects the microRNA-146b/SIRT1 and AMPK/SIRT1 pathways in ethanol-induced liver injury (*Zhang et al., 2015b*). Cryptotanshinone and Korean Red Ginseng activate the AMPK/SIRT1 pathway, contributing to ALD treatment (*Nagappan et al., 2020*; *Han et al., 2015*). Berberine ameliorates lipid metabolism abnormalities and liver injury *via* the AMPK/SIRT1 pathway in AFLD mice (*Zhu et al., 2023*).

### Autophagy and cellular homeostasis regulators

**Salvianolic acid A** Influences autophagosome-lysosome fusion, essential for cellular homeostasis in liver disease (*Shi et al., 2018*). Carvacrol and Cilostazol combination demonstrate hepatoprotective effect against ALF through the SIRT1/Nrf2/HO-1 pathway, exhibiting antioxidant, anti-inflammatory, and anti-fibrotic features (*Abu-Risha et al., 2023*).

### Anti-inflammatory and histone modification agents

**Therapeutic butein administration** prevents ethanol-associated acetylation changes in histones, reducing CCL2 expression and hepatic inflammation (*Ghare et al., 2023*). Botulin also known for activating the AMPK/SIRT1 pathway, broader implications on inflammatory processes in ALD are noteworthy (*Bai et al., 2016b*).

*Drugs for NAFLD.* The drugs under investigation have demonstrated their potential in reducing adipogenesis through the activation of AMPK. Salvianolic acid A (*Li et al., 2020a*), lycium barbarum polysaccharide (*Jia et al., 2016*), and atractylenolide III (*Li et al., 2022b*) have been shown to increase AMPK levels, consequently inducing SIRT1 expression in high-fat diet-induced mice. Berberine, on the other hand, functions as a miR-146b inhibitor, leading to increased SIRT1 expression and subsequent deacetylation of FoxO1,

thus ameliorating hepatic insulin sensitivity (*Sui et al., 2021*). Furthermore, salvianolic acid B has been found to inhibit HMGB1 *via* SIRT1, effectively suppressing inflammation in NAFLD (*Zeng et al., 2015*). Silymarin has also exhibited hepatoprotective effects in NAFLD mice through its interaction with SIRT1 (*Chang et al., 2019*).

**AMPK pathway and the lipid accumulation**

Salvianolic Acid A, Lycium Barbarum Polysaccharide, Atractylenolide III, Tetrahydropalmatine increase AMPK levels, leading to the induction of SIRT1 expression (*Li et al., 2020a*; *Jia et al., 2016*; *Li et al., 2022b*; *Yin, Liu & Wang, 2023*). This pathway is especially effective in high-fat diet-induced NAFLD models. By enhancing AMPK and SIRT1 activity, they help reduce adipogenesis, crucial in NAFLD progression. Kaempferol also regulates hepatic lipid accumulation through SIRT1/AMPK signaling (*Li et al., 2023*).

**Others**

Berberine acts as a miR-146b inhibitor, increasing SIRT1 expression and deacetylating FoxO1 and ameliorates hepatic insulin sensitivity, a crucial aspect in NAFLD management (*Sui et al., 2021*). Salvianolic Acid B inhibits HMGB1 *via* SIRT1, suppressing inflammation in NAFLD (*Zeng et al., 2015*). Silymarin exhibits hepatoprotective effects in NAFLD *via* SIRT1 interaction (*Chang et al., 2019*). 4-Butyl-Polyhydroxybenzophenone promotes mitochondrial biogenesis and prevent NAFLD liver injury by activating the PGC1 $\alpha$ pathway in a SIRT1-dependent manner (*Song et al., 2023*).

*Treatment for IRI.* While options for managing IRI in liver diseases are limited, several treatments have emerged that effectively protect hepatocytes. Trimetazidine in the IGL-1 solution inhibits p53 and activates SIRT1, thus protecting hepatocytes from IRI (*Pantazi et al., 2015b*). Rebamipide reduces inflammation and improves histological alterations in the liver, crucial for recovery post-reperfusion, which increases SIRT1 expression and modulates$\beta$-catenin and FOXO1, while suppressing NF-$\kappa$B p65 expression (*Gendy, Abdallah & El-Abhar, 2017*; *Elwany et al., 2023*). Losartan mitigates liver injury associated with transplantation, improving overall transplantation success rates *via* SIRT1, enhancing the liver's defense against oxidative stress and inflammation (*Pantazi et al., 2015a*). Tauroursodeoxycholic acid modulates the SIRT1/FXR signaling pathway beneficial in managing IRI during liver surgeries (*Sun et al., 2020*). Pachymic acid maintains SIRT1 expression during oxygen-glucose deprivation/reoxygenation (*Xue et al., 2023a*; *Xue et al., 2023b*).

*Drugs for cholestatic liver injury.*

**SIRT1/FXR pathway modulators**

In the realm of cholestatic liver disease treatment, SIRT1/FXR pathway modulators have emerged as key therapeutic agents. Baicalin (*Yang et al., 2020a*) and Celastrol (*Zhao et al., 2019*) effectively modulate this pathway, offering hepatoprotective benefits. Similarly, chlorogenic acid acts by facilitating SIRT1-mediated deacetylation of FXR, thus mitigating hepatotoxicity (*Zhu et al., 2018b*). Additionally, liquiritin plays a multifaceted role by regulating both the Sirt1/FXR/Nrf2 pathway and bile acid transporters, leading to reduced hepatotoxicity and cholestasis (*Yan et al., 2023*).

**SIRT1/AMPK/Nrf2 pathway regulators**

Ginsenoside Rg1 have demonstrated effectiveness in alleviating cholestatic liver injury by regulating the SIRT1/AMPK/Nrf2 pathway, crucial in reducing oxidative stress (*Gao et al., 2023*). Similarly, Salvianolic Acid A plays a significant role in managing oxidative stress and liver injury through its involvement in SIRT1 regulation and the participation of HSF1 (*Zhu et al., 2018a*).

*Drugs for fibrosis.* The key approach in treating liver fibrosis involves preventing the activation of HSCs, as their activation is a pivotal factor in the development of fibrosis.

**Activation of HSC**

Dioscin directly inhibits the activation of HSCs, preventing them from transforming into myofibroblast-like cells that produce fibrotic tissue (*Gu et al., 2016*). Ampelopsin induces autophagy in HSCs, which reduces their activation and subsequent fibrosis progression (*Ma et al., 2019*). Curcumol alleviates endoplasmic reticulum stress in HSCs (*Sun et al., 2022*). This reduction in stress is critical as ER stress is known to contribute to HSC activation and the advancement of fibrosis.

**Other mechanisms**

Gardenia Fructus upregulates the AMPK/SIRT1 pathway (*Shin et al., 2021*). This modulation controls factors leading to HSC activation and fibrosis, including managing cellular energy metabolism and reducing inflammation and oxidative stress. Cichorium Intybus Linn modulates the SIRT1/FXR signaling pathway (*Keshk et al., 2019*).

***Special intervention***

Several therapeutic interventions have been identified as effective approaches in alleviating liver injury through the modulation of SIRT1. These interventions include hypothermic machine perfusion (*Zeng et al., 2017*), treatment with partially hydrolyzed guar gum (*Liu et al., 2019b*), ischemic preconditioning (*Pantazi et al., 2014*), the combined administration of blueberry juice and probiotics (*Zhu et al., 2016*; *Fan et al., 2021*), as well as carbon monoxide therapy (*Sun et al., 2017*; *Kim et al., 2015*).

In conclusion, targeting SIRT1 in liver injury represents a comprehensive approach, addressing different aspects of liver disease pathology from inflammation in ALI, metabolic dysregulation in ALD and NAFLD, oxidative stress in IRI, to HSC activation in fibrosis. The versatility of SIRT1 as a therapeutic target underscores its potential in providing effective treatments across a spectrum of liver conditions. Drug repurposing further enhances this potential by offering a pool of existing drugs for possible reapplication in liver disease therapy. Some special intervention may provide a novel way for liver diseases.

## CONCLUSION

In this review, we have delved into the multifaceted nature of SIRT1 in liver pathology, exploring its diverse biological characteristics and therapeutic implications. The variable expression of SIRT1 in hepatocytes, HSCs, and Kupffer cells, and its distinct roles in various liver diseases such as ALI, ALD, NAFLD, IRI, cholestatic injury, and liver fibrosis, underscore its complex function in hepatology. Intriguingly, SIRT1 exhibits different

characteristics and functions depending on the type of liver disease, revealing a level of adaptability that is not yet fully understood.

The potential of SIRT1 as a therapeutic target is further highlighted by the effectiveness of various drugs, especially components of TCMs, in ameliorating liver injury. The hepatoprotective effects of certain foods, medicinal diets, and diet therapies also point towards natural therapies as promising avenues for liver disease treatment. However, the current scarcity of effective treatments for liver injury emphasizes the urgent need for more specific and targeted therapeutic strategies.

Despite the established role of SIRT1 in liver disease, many aspects remain unclear, particularly its adaptive function. The complex interactions of SIRT1 with other biomolecules in the liver are not fully elucidated. Contradictory findings, such as the simultaneous induction of p-AMPK and suppression of SIRT1, and instances where SIRT1 exacerbates liver injury by inflammation, present challenges in understanding its selective function in different disease contexts. These discrepancies underscore the need for further research to unravel the intricate mechanisms of SIRT1's action in liver diseases.

To advance our understanding and therapeutic potential, future research must focus on clarifying the adaptive function of SIRT1, resolving contradictory findings, and exploring the full spectrum of its roles, beyond deacetylation, in liver pathology. Such insights will be crucial in harnessing SIRT1's full potential as a novel and effective therapeutic target for liver diseases.

### Funding

This research was financially supported by the National Natural Science Foundation of China (32260089), the Subject of Innovation and Entrepreneurship Education of Guizhou Ordinary Undergraduate Colleges (2022SCJZW10), the Future Outstanding Teachers Training Program of Zunyi Medical University (ZMUWLJXMS-2021XDL), the Postgraduate Teaching Reform Project of Zunyi Medical University (ZYK105), the Joint Biding Project of Zunyi Science & Technology Department and Zunyi Medical University (No. ZSKHHZ[2020]91), and the Science and Technology Department Foundation of Guizhou Province of China (No. QKPTRC [2019]-027). The funders had no role in study design, data collection and analysis, decision to publish, or preparation of the manuscript.

### Grant Disclosures

The following grant information was disclosed by the authors:
National Natural Science Foundation of China: 32260089.
Subject of Innovation and Entrepreneurship Education of Guizhou Ordinary Undergraduate Colleges: 2022SCJZW10.
Future Outstanding Teachers Training Program of Zunyi Medical University: ZMUWLJXMS-2021XDL.
Postgraduate Teaching Reform Project of Zunyi Medical University: ZYK105.

Joint Biding Project of Zunyi Science & Technology Department and Zunyi Medical University: ZSKHHZ[2020]91.
Science and Technology Department Foundation of Guizhou Province of China: QKPTRC [2019]-027.

## Competing Interests

The authors declare there are no competing interests.

## Author Contributions

- Mufei Wang conceived and designed the experiments, performed the experiments, analyzed the data, prepared figures and/or tables, authored or reviewed drafts of the article, and approved the final draft.
- Juanjuan Zhao conceived and designed the experiments, prepared figures and/or tables, authored or reviewed drafts of the article, and approved the final draft.
- Jiuxia Chen performed the experiments, prepared figures and/or tables, authored or reviewed drafts of the article, and approved the final draft.
- Teng Long performed the experiments, analyzed the data, authored or reviewed drafts of the article, and approved the final draft.
- Mengwei Xu analyzed the data, authored or reviewed drafts of the article, and approved the final draft.
- Tingting Luo analyzed the data, authored or reviewed drafts of the article, and approved the final draft.
- Qingya Che analyzed the data, authored or reviewed drafts of the article, and approved the final draft.
- Yihuai He conceived and designed the experiments, prepared figures and/or tables, authored or reviewed drafts of the article, and approved the final draft.
- Delin Xu conceived and designed the experiments, analyzed the data, prepared figures and/or tables, authored or reviewed drafts of the article, and approved the final draft.

## Data Availability

This is a literature review.

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
