# Peer review of "The role of sirtuin1 in liver injury: molecular mechanisms and novel therapeutic target"

_PeerJ, doi:10.7717/peerj.17094_

## Round 0.1 · original submission · Major Revisions

Dear Dr. Xu,

Thank you for your patience while your manuscript was peer-reviewed at PeerJ. It has now been evaluated by the editor and by three independent reviewers with relevant expertise.

All the reviewers have concerns about the conclusion part. The reviewers also provided some suggestions to improve the manuscript. I have thoroughly read the manuscript and feel English expression should be improved in a concise way, which overlap with the comments made by the other reviewers.

I invite you to respond to the reviewers' detailed comments and revise your manuscript. All the reviewers' comments need to be addressed before the manuscript can be accepted.

Thank you for submitting your manuscript to PeerJ and I look forward to receiving your revision.

Best,

Xiaotian Tang, PhD
Academic Editor, PeerJ
[email protected]

**Language Note:** The Academic Editor has identified that the English language must be improved. PeerJ can provide language editing services - please contact us at [email protected] for pricing (be sure to provide your manuscript number and title). Alternatively, you should make your own arrangements to improve the language quality and provide details in your response letter. – PeerJ Staff

Reviewer 1 ·

Basic reporting

no comment

Experimental design

no comment

Validity of the findings

no comment

Additional comments

General comments:
Liver disease is a prevalent and critical menace to human health. This article presents a groundbreaking and optimistic molecular target, SIRT1, for liver diseases in terms of both mechanism and treatment. It is reasonable to clarify the roles of SIRT1 in six type of liver injury and discuss the therapy targeting to SIRT1. However, there are several inadequacies throughout the manuscript that necessitate thorough examination and revision. Certain aspects in the article require additional enhancement. In general, this article is suitable for publication in this journal, but it must be revised prior to its release.
The summary of chapter 1 and 2 could be improved. It may be better to refine some scientific questions and propose further research opinions.
The conclusions and perspectives are not well structed, which means the paragraphs are not coherent enough, and the inference is not logical enough. The authors should emphasize the importance of SIRT1 in liver injury, and further put forward the views on the research of mechanism of SIRT1 in liver injury and targeted therapy. And this can better reflect the value of this article.
The exposition of mechanism of SIRT1 in liver injury is well, however, there are still some deficiencies in detail, which are listed below:
Specific comments:
(1) The meaning of the sentence in lines 45 to 46 does not match with Figure 1.
(2) The expression "SIRT1's" in line 56 is not proper language.
(3) The time description seems to be incorrect in line 64.
(4) Sentences in lines 77 to 80 are incoherent and need to be restructured.
(5) The description in lines 456 to 457 is unclear. It is not specified which cell experiences these changes within the context.
(6) The word "comprehensive" is repeated in lines 503 to 504.

Annotated reviews are not available for download in order to protect the identity of reviewers who chose to remain anonymous.

Reviewer 2 ·

Basic reporting

This article will be of interest to readers or researchers of those interested in novel targets to liver injury or mechanism of new molecules. SIRT1 belong to the class III histone deacetylase family of proteins are crucial for various biochemical processes within cells, which seems to be a hot topic in liver disease.

Experimental design

The present review contained the researches with original studies and reviews in English since 2018 from Web of Science, Google Scholar database and PubMed. The keywords used SIRT1, liver injury, liver fibrosis, liver ischemic, NAFLD and ALD, which is enough to explain the rigor of this study.

Validity of the findings

This review outlines the key signaling pathways associated with sirtuin1 and liver injury, and discusses recent advances in therapeutic strategies targeting sirtuin1 in liver diseases.

Additional comments

In my opinion, it appears to be well designed, but the following points need to be modified to make the manuscript better:
1.It would be better if the summary paragraphs of part 1 and 2 are more concise and prominent in views.
2.There are some inappropriate English expressions that need to be corrected.
3.In the “Conclusion” part, authors mentioned HCC but less summaries in liver injures described above. Maybe authors can expand on the connection between HCC and liver injures, or emphasize the main idea of the article more prominent.
4.Some details need to be corrected as following:
(1)Additional references are needed for the sentence in lines 100 to 103.
(2)It is worth to noticing the common ground of the therapies target to SIRT1 in the part 2.
(3)The paragraph in lines 456 to 457 is confused in the context, in which the cellular processes need to be clarified to match to adequate cells.
(4)Is there a format error in lines 633 to 635?

Reviewer 3 ·

Basic reporting

In this study, provided a comprehensive summary of the biological characteristics of SIRT1 and related therapeutic applications in liver injury, including acute liver injury, alcoholic liver disease, nonalcoholic fatty liver disease, ischemia-reperfusion injury, cholestatic injury and liver fibrosis. It is possibly providing a great resource for the future research and inspire novel clinical approaches to treating liver injuries.

Experimental design

1. When discussing the SIRT1 in NAFLD, you included ‘upstream regulation of SIRT1, SIRT1 in lipid metabolism and production, and SRIT1 in other processes contributing NAFLD.’ Are these three items independent from each? Shouldn`t SIRT1 has upstream regulator in either lipid metabolism or other processes involved in NAFLD?
2. People would care more about how SIRT1 inhibit IRI instead of the process of oxidative injury and apoptosis in IRI, part 1.4.1 doesn’t help too much.

Validity of the findings

Conclusion part is weak. Why SIRT3 is included while the whole paper doesn`t talk anything related to other SIRT family at all? The unresolved questions/gaps/future directions should be described clearer for the SIRT1 in liver injury.

Annotated reviews are not available for download in order to protect the identity of reviewers who chose to remain anonymous.

---

## Round 0.2 · Minor Revisions

Dear authors, thank you for your submission to PeerJ and for your patience. I took over the submission replacing Dr Xiaotian Tang. I decided to invite other reviewers as it is my perspective that your work does not only lack novelty but neither presents a Critical perspective nor even clear molecular and cellular arguments to constitute a new body of literature. A reviewer managed to submit a timely review. Please, check it. Additionally, please introduce what is your motivation, knowledge gap, or criticism of the current field status since the role of sirtuin in hepatology is well established.

Reviewer 1 ·

Basic reporting

no comment

Experimental design

no comment

Validity of the findings

no comment

Additional comments

no comment

Reviewer 2 ·

Basic reporting

Yes.

Experimental design

Yes.

Validity of the findings

Yes.

Reviewer 3 ·

Basic reporting

No comment

Experimental design

No comment

Validity of the findings

No comment

Additional comments

No comment

Reviewer 4 ·

Basic reporting

no comment

Experimental design

no comment

Validity of the findings

no comment

Additional comments

The manuscript is clearly written in professional and unambiguous language. If there is one weakness, it would be the need to pay attention to the accuracy of the reference citation format. Improvements can be made, for example, on line 692 and other similar instances.

·

Basic reporting

Abstract

Please include this is a narrative review. Delete the last sentence.

Introduction

Replace in this paper by in this manuscript.
The introduction must be rewritten to group together SIRT information. You must not mention a figure, you must describe it. In addition, in the legend you must synthesize the information.

Concerning the structure of the text, it needs to be completely reworked so as to group information by mechanism (inflammation, apoptosis, oxidative stress, epigenetic regulation, etc.) and not by "pathology", as this is very redundant to read.
also synthetize your ideas in the same paragraph, you indicate the miRNA in several places and you make a paragraph on it (1.5.4 Others ).

Figures

As mentioned, the legend of each figure must be accompanied by a short text describing it.

Experimental design

The structure of the review must be improved.

Validity of the findings

Explore SIRT-1 in the liver diseases is very interesting, with the increase in chronic liver disease, notably NASH, but to date the manuscript does not sufficiently highlight this potentially relevant information.
Natural or pharmacological therapies are also not sufficiently highlighted.

---

## Round 0.3 · accepted · Accept

Dear authors, I am happy to let you know that your manuscript will now be moving forward. Thank you for your diligent work.